# Innovative Insights for Establishing a Synbiotic Relationship with *Bacillus coagulans*: Viability, Bioactivity, and *In Vitro*-Simulated Gastrointestinal Digestion

**DOI:** 10.3390/foods12193692

**Published:** 2023-10-08

**Authors:** Saranya Suwanangul, Pannapapol Jaichakan, Nukrob Narkprasom, Supaluck Kraithong, Kanjana Narkprasom, Papungkorn Sangsawad

**Affiliations:** 1Program in Food Science and Technology, Faculty of Engineering and Agro-Industry, Maejo University, Chiang Mai 50290, Thailand; saranya_sw@mju.ac.th; 2Faculty of Business Administration, Chitralada Technology Institute, Bangkok 10300, Thailand; pannapapol.jai@cdti.ac.th; 3Program in Food Engineering, Faculty of Engineering and Agro-Industry, Maejo University, Chiang Mai 50290, Thailand; narkprasom@gmail.com; 4Guangdong Provincial Key Laboratory of Food Quality and Safety, College of Food Science, South China Agricultural University, Guangzhou 510642, China; supaluck@scau.edu.cn; 5School of Animal Technology and Innovation, Institute of Agricultural Technology, Suranaree University of Technology, Nakhon Ratchasima 30000, Thailand

**Keywords:** antidiabetic, antioxidative activity, *Bacillus coagulans* spores, microencapsulation, viability

## Abstract

This study investigates the use of encapsulating agents for establishing a synbiotic relationship with *Bacillus coagulans* (TISTR 1447). Various ratios of wall materials, such as skim milk powder, maltodextrin, and cellulose acetate phthalate (represented as SMC1, SMC3, SMC5, and SMC7), were examined. In all formulations, 5% inulin was included as a prebiotic. The research assessed their impact on cell viability and bioactive properties during both the spray-drying process and in vitro gastrointestinal digestion. The results demonstrate that these encapsulating agents efficiently protect *B. coagulans* spores during the spray-drying process, resulting in spore viability exceeding 6 log CFU/g. Notably, SMC5 and SMC7 displayed the highest spore viability values. Moreover, SMC5 showcased the most notable antioxidant activity, encompassing DPPH, hydroxy radical, and superoxide radical scavenging, as well as significant antidiabetic effects via the inhibition of α-amylase and α-glucosidase. Furthermore, during the simulated gastrointestinal digestion, both SMC5 and SMC7 exhibited a slight reduction in spore viability over the 6 h simulation. Consequently, SMC5 was identified as the optimal condition for synbiotic production, offering protection to *B. coagulans* spores during microencapsulation and gastrointestinal digestion while maintaining bioactive properties post-encapsulation. Synbiotic microcapsules containing SMC5 showcased a remarkable positive impact, suggesting its potential as an advanced food delivery system and a functional ingredient for various food products.

## 1. Introduction

Creating functional foods via the enrichment of novel foods with probiotic organisms or spores is a promising approach. Processed foods with improved health benefits compared to traditional nutritional options can now be produced using the advancements in current food processing technology [1]. In contemporary society, consumers are growing increasingly health-conscious, giving greater significance to the nutritional value of their food choices. Consequently, the successful acceptance of novel foods depends on the concept of food quality and additional value-added food functionalities, particularly those related to probiotics [2].

Probiotics are academically defined as living microorganisms that, when administered in sufficient quantities, bestow advantageous health effects upon the host organism [3]. Products must contain a minimum of 10^6^ CFU/g of probiotic microorganisms and ensure the preservation of their viability throughout their designated shelf life to attain probiotic status and access these benefits [4]. *Bacillus coagulans* (*B. coagulans*) find wide application in food and feed additives due to their effective inhibition of harmful bacterial growth. Nonetheless, a significant proportion of these microorganisms lose their activity during the manufacturing, storage, and rigorous processing associated with functional foods, primarily owing to their susceptibility to stomach acid and elevated temperatures [5]. Altun and Erginkaya [6] provide evidence that *B. coagulans* can thrive within a pH range of 5.5–6.2 and release spores at 37 °C, leading to its designation as Generally Recognized as Safe (GRAS). In the small intestine, this bacterium’s spores assume a crucial role in the digestion of carbohydrates within the human gastrointestinal tract [7]. Probiotics fulfill a crucial role in maintaining the balance of the gastrointestinal system and energy levels via the production and regulation of essential metabolites, encompassing enzymes, vitamins, peptides, essential amino acids, and antioxidants [8,9,10]. These metabolites are vital for promoting human well-being. Probiotics offer an array of significant benefits, including the prevention of conditions such as hypertension [11] and diabetes [12], a reduction in cholesterol levels [13], displaying anti-carcinogenic properties [14], and the alleviation of conditions like anxiety, depression [15], and allergies. Furthermore, probiotics actively modulate the immune system [16] and provide protection against food allergies and atopic diseases [17], making them an indispensable component of comprehensive health management. Microencapsulation is a technique that provides physical protection to bacterial spores and bioactive compounds, effectively shielding them from chemical degradation and preserving their functionality, especially in the context of the food industry. For instance, bioactive peptides produced from bacteria such as *Phaseolus lunatus* can be safeguarded from the harsh gastrointestinal environment by microencapsulation using materials like maltodextrin/gum Arabic [18]. This evolving method holds promise for the preservation of microbial isolates. Furthermore, encapsulation holds the potential to enable controlled release and optimize delivery to specific target sites, thereby enhancing the overall efficacy of probiotics. Nonetheless, ensuring the viability of probiotic-containing products remains a substantial challenge. Probiotics must withstand the rigors of industrial processing, various storage conditions, and the expedition through the gastrointestinal tract. Notably, microencapsulation technology has proven effective in safeguarding probiotic cells and spores, significantly bolstering their viability within food products and throughout the gastrointestinal tract [19]. Encapsulation serves as a valuable means for separating probiotics from food matrices. However, its success is contingent upon several critical factors, including the choice of coating materials, the specific encapsulation technology applied, storage conditions, capsule quality, and the sensitivity of the probiotic strain. In essence, encapsulation has the potential to efficiently isolate probiotics from food, but its actual effectiveness hinges on the considerations outlined above [20].

The extensive application of microencapsulation technology in combination with thermal protection agents has been underscored in a study by Misra et al. [21]. Among various methods, spray drying stands out as a preferred choice for probiotics due to its cost-effectiveness and high-yield efficiency. However, challenges arise, as seen in the work of Yao et al. [22] where *Lactobacillus plantarum* (*L. plantarum*) 550 lost viability during this process. To enhance cell survival, proteins and polysaccharides are often added to bacteria to create probiotic microcapsules. Skim milk powder, due to its low viscosity similar to calcium alginate, is a valuable encapsulating agent. It effectively enhances the survival of encapsulated materials in the harsh lower gastric environment compared with free cells [23]. Proteins are commonly employed as wall materials in the spray-drying process to form protective layers around cells swiftly, safeguarding probiotic bacteria from thermal stress [24]. Maltodextrin is favored as a carrier in spray-dried hygroscopic powders, facilitating interactions with liquids and solids, making it a versatile choice for incorporating various substances [25]. Additionally, cellulose acetate phthalate functions as an effective polymer for pH-controlled release, dissolving in mildly acidic or neutral intestinal conditions while withstanding exposure to highly acidic gastric fluids [26]. Researchers have explored multiple wall matrix systems, including ternary wall matrices, to enhance probiotics’ viability and physicochemical characteristics [27,28]. Nonetheless, there is a notable gap in the literature regarding specific combinations of encapsulation materials, such as skim milk powder, maltodextrin, and cellulose acetate phthalate, for safeguarding *B. coagulans* spores during the intricate spray-drying process.

As far as current knowledge extends, there is a notable lack of research on the combination of a synbiotic formulation involving *B. coagulans* spores with specific encapsulation materials like skim milk powder, maltodextrin, and cellulose acetate phthalate while also incorporating the prebiotic inulin. Therefore, the primary aim of this study is to investigate how different encapsulation materials affect the viability of probiotic *B. coagulans* spores. Furthermore, the research seeks to assess the multifaceted bioactive properties of these spores, including their antioxidant and antidiabetic activities, which are integral to health promotion, especially in the context of the spray-drying process. Additionally, this study explores the resilience of *B. coagulans* spores during simulated gastrointestinal digestion.

## 2. Materials and Methods

### 2.1. Materials

The probiotic culture of *B. coagulans* (TISTR 1447) was sourced from the Thailand Institute of Scientific and Technological Research (TISTR) for our research conducted at Maejo University’s Food Safety and Biotechnology Lab. Gastrointestinal fluids were simulated using pepsin porcine (P6887) and pancreatin (P7545) from Sigma-Aldrich Co. in the St. Louis, MI, USA. In this research project, we utilized chemicals of analytical reagent-grade quality.

### 2.2. Production of Spore Suspensions

The production followed the methodology outlined by Russell et al. [29] with minor adjustments. The first step involved initiating sporulation in *B. coagulans* TISTR 1447 by cultivating cells in Nutrient Broth at 37 °C for 24 h. Subsequently, there was aerobic growth on Nutrient Yeast Extract Salt Medium (NYSM) agar for 24 h at the same temperature. In the second step, culturing was carried out in NYSM broth using a single colony from the agar plate. This culture was shaken at 250 rpm and maintained at 37 °C for five days, resulting in a 90% sporulation rate. Bacterial sediment was successfully obtained by subjecting the bacterial suspension to centrifugation at 4000× *g* for 10 min, followed by a washing step and resuspension in 0.9% Sodium Chloride Sterile Saline, totaling one hundred microliters. To effectively eliminate vegetative bacterial forms, a heat sterilization process was employed (80 °C for 15 min). Ultimately, the original stock solution underwent dilution, decreasing its concentration from 1 × 10^12^ CFU/g to a final concentration of 1 × 10^10^ CFU/g. This thinned solution was subsequently stored under refrigeration and prepared for future use, as depicted in Figure 1.

### 2.3. Production of Synbiotic Microcapsules by Spray Drying

The preparation of microencapsulation beads adhered to the methodology delineated by Arslan-Tontul and Erbas [30], albeit with minor adaptations. In this process, the encapsulation of *B. coagulans* spores within a probiotic culture was executed by dissolving these spores within a 200 mL aqueous solution. This solution was composed of a mixture of skim milk powder, maltodextrin, and cellulose acetate phthalate in different ratios, precisely defined as 1:2:1, 3:2:1, 5:2:1, and 7:2:1. Additionally, it was fortified with 5% (*w*/*v*) inulin, a well-established prebiotic. Subsequently, the entire mixture underwent sterilization via a heating regimen at 121 °C for a duration of 15 min, thereby yielding synbiotic microcapsules of distinct formulations, designated as SMC1, SMC3, SMC5, and SMC7. After the solutions were allowed to cool, *B. coagulans* spores were introduced into the mixture in a proportion of 20% *v*/*w* relative to the mass of the wall materials. This introduction maintained an initial cell count of 1 × 10^10^ CFU/g within the microencapsulation process. These inoculated solutions underwent microencapsulation using a laboratory-scale spray dryer, specifically the Buchi-290 from Switzerland. Temperature control was maintained, with an inlet temperature set at 120 °C and an outlet temperature at 50 °C. In a two-fluid nozzle system, compressed air at 0.3 bar is used to aspirate, and the feeding rate of the solutions is approximately 16.5 mL/min. A cyclone separated the synbiotic microcapsules, accumulating them in a designated vessel. The collected microcapsules were then preserved at −18 °C for subsequent analysis (Figure 2). The survival rate (%) in comparison to the initial count prior to spray drying was computed following the approach outlined by Gomez-Mascaraque et al. [31].

(1)
$$\mathrm{Survival}\;\mathrm{rate}\;(\%)=\frac{N(F)}{N(0)}\times 100$$


The equation value was determined by evaluating the probiotic cell count (log CFU/g of dry solid material) both prior to and following the spray-drying process. In this context, N(F) denotes the count of probiotic cells after the spray-drying process, while N(0) signifies the count of probiotic cells before the spray-drying procedure.

### 2.4. In Vitro Bioactive Properties of Synbiotic Microcapsules

We conducted an analysis of the spray-dried samples, comprising free cells, unencapsulated cells, and encapsulated cells, to assess their bioactive properties.

#### 2.4.1. 1,1-Diphenyl-2-Picrylhydrazyl (DPPH) Free-Radical Scavenging

The evaluation of DPPH free-radical-scavenging activity followed a previously established method by Yan et al. [10], with slight modifications to adapt it for the microplate technique. The spray-dried sample was dissolved in 0.1 M sodium phosphate buffer with a pH of 7.0, yielding a concentration range of 0–20 mg solid/mL. In the initial step, one hundred microliters of the sample or buffer (utilized as a positive control) was mixed with an equal volume of an ethanolic DPPH radical solution at a concentration of 0.2 mM or absolute ethanol (used as a blank). Following this, in the second step, the mixture was left to incubate at room temperature in the absence of light for 30 min, after which the absorbance was measured at 517 nm. The DPPH radical-scavenging activity was calculated using the following equation:
(2)
$$\mathrm{DPPH}\;\mathrm{radical}-\mathrm{scavenging}\;\mathrm{activity}\;(\%)=1-\frac{A\;sample-A\;blank}{A\;control}\times 100$$


#### 2.4.2. Hydroxyl Radical Scavenging (HRSA)

The assessment of HRSA (Hydroxyl Radical-Scavenging Activity) was carried out according to a previously established protocol by Suwanangul et al. [32], with minor modifications to adapt the method for microplate analysis. The spray-dried sample was dissolved in 0.1 M sodium phosphate buffer at pH 7.4, achieving a concentration range of 0–20 mg solid/mL. In the first step, the reaction mixture was prepared that included a 3 mM solution of 1,10-phenanthroline in water (50 µL), 0.1 M sodium phosphate buffer at pH 7.4 (50 µL), the samples (50 µL), and 3 mM FeSO_4_ (50 µL) in a 96-well microplate. Subsequently, in the second step, a 20 mM hydrogen peroxide solution (50 µL) was added to initiate the reaction. This was followed by incubation at 37 °C for 90 min. Afterward, the absorbance at 517 nm was measured, and the determination of hydroxyl radical-scavenging ability was calculated according to the procedure outlined below:
(3)
$$\mathrm{HRSA}\;(\%)=\frac{\left(A\;samples-A\;blank\right)}{\left(A\;control-A\;blank\right)}\times 100$$


#### 2.4.3. Superoxide Radical Scavenging

The determination of superoxide radical-scavenging ability (SRSA) followed the method outlined by Yan, Li, Yue, Wang, Zhao, Evivie, Li, and Huo [10], with slight adaptations to accommodate a 96-well clear flat-bottom plate. Initially, 80 µL of the spray-dried sample was dissolved in 50 mM Tris–HCl solution at pH 8.2, with a concentration range of 0–20 mg solid/mL. Alternatively, deionized water was used for the control group and was combined with 80 µL of 50 mM Tris-HCl solution. Following this, the resulting mixture was incubated at 25 °C for 20 min. An amount of 40 µL of 25 mM pyrogallol in 10 mM HCl was introduced, and the mixture was left to stand at room temperature for 4 min. The determination of superoxide anion radical-scavenging activity was carried out as described below:
(4)
$$\mathrm{SRSA}\;(\%)=1-\frac{A\;samples}{A\;control}\times 100$$


The results have been presented as EC_50_ values, which indicate the sample concentrations needed to scavenge 50% of DPPH, HRSA, and SRSA. These EC_50_ values were determined by utilizing a linear regression curve derived from a range of the spray-dried sample concentrations from 0 to 20 mg/mL.

#### 2.4.4. In Vitro Activities of α-Glucosidase

The determination of α-glucosidase-inhibitory activity followed a method previously outlined by Zhang et al. [33], with slight modifications. The spray-dried sample was dissolved in 50 mM Tris–HCl solution at pH 8.2, with a concentration range of 0–20 mg solid/mL. Alternatively, it was solubilized in 0.02 M sodium phosphate buffer at pH 6.9, with a concentration range of 0–40 mg solid/mL. In summary, a mixture of 50 µL of our samples and 50 µL of a 10 mM PNP-glycoside solution (dissolved in 0.1 M sodium phosphate buffer at pH 6.9) was prepared within a 96-well microplate. Subsequently, 50 µL of 0.2 U/mL α-glucosidase enzyme solution was added to initiate the reaction. The microplate was then placed in an incubator set at 37 °C for 30 min. The release of p-nitrophenol from PNP-glycoside was quantified by measuring the absorbance at 405 nm using a microplate reader (Thermo Scientific™, Waltham, MA, USA). Acarbose was evaluated using the same procedure and served as our positive control. The extent of α-glucosidase-inhibitory activity was determined as follows:
(5)
$$\alpha -\mathrm{glucosidase}\;\mathrm{inhibition}\;(\%)=\frac{\left(A\;control-A\;control\;blank\right)-(A\;sample-A\;sample\;blank)}{\left(A\;control-A\;control\;blank\right)}\times 100$$


#### 2.4.5. In Vitro Activities of α-Amylase

The determination of α-amylase-inhibitory activity was conducted following a method previously documented by Mudgil et al. [34], with some modifications. In this adapted approach, p-nitrophenyl α-D maltohexaoside (pNPM; 5 mmol/L) served as the substrate. The procedure was outlined by preparing a mixture consisting of 50 µL of pNPM, 50 µL of our samples (which were dissolved in 0.02 M sodium phosphate buffer at pH 6.9 with a concentration range of 0–40 mg solid/mL), and 50 µL of the PPA enzyme solution (50 mg/mL) in sodium phosphate buffer (0.02 M, pH 6.9). This mixture was then incubated for 30 min at 37 °C. Subsequently, the release of p-nitrophenol at 405 nm was monitored. The extent of α-amylase-inhibitory activity was calculated as follows:
(6)
$$\alpha -\mathrm{amylase}\;\mathrm{inhibition}\;(\%)=1-\frac{\left(A1-A2\right)}{\left(A3-A4\right)}\times 100$$


In this equation, A1 signifies the absorbance measured in the reaction wells containing both the enzyme and the test sample, whereas A2 stands for the absorbance obtained in the reaction blank containing only the test sample without the enzyme. A3 represents the absorbance recorded in the control wells with the enzyme but without the test sample, and finally, A4 indicates the absorbance measured in the control blank, which lacks both the enzyme and the test sample.

#### 2.4.6. Dipeptidyl Peptidase IV (DPP-IV) Inhibition

The assessment of DPP-IV-inhibitory activity followed a method described by Sangsawad et al. [35], with minor adjustments. The spray-dried sample was dissolved in 50 mM Tris–HCl solution at pH 8.2, achieving a concentration range of 0–20 mg solid/mL. Alternatively, it was solubilized in 100 mM Tris-HCl buffer at pH 8.0, with a concentration range of 0–40 mg solid/mL. Specifically, a twenty-five microliter aliquot of the diluted sample (diluted in 100 mM Tris-HCl buffer at pH 8.0) was mixed with twenty-five microliters of the prepared 1.6 mM Gly-Pro-p-nitroanilide substrate. The sample–substrate mixture was pre-incubated for 10 min at 37 °C. Following this, fifty microliters of DPP-IV (0.01 U/mL) were added, and the reaction proceeded for 60 min at 37 °C. To halt the reaction, researchers introduced one hundred microliters of 1.0 M sodium acetate buffer at pH 4.0 and measured the absorbance at 405 nm. The absorbance values were then normalized against sample blanks where DPP-IV was replaced with Tris-HCl buffer. Negative controls (representing no DPP-IV activity) and positive controls (indicating DPP-IV activity without an inhibitor) were prepared by substituting the sample and DPP-IV solution with Tris-HCl buffer, respectively. The DPP-IV inhibition rate for each sample was calculated as follows:

(7)
$$\mathrm{DPP}-\mathrm{IV}\;\mathrm{inhibition}\;(\%)=1-\frac{\left(A\;sample-A\;sample\;blank\right)}{\left(A\;positive\;control-A\;negative\;control\right)}\times 100$$


The IC_50_ values for each sample were determined via curve interpolation, indicating the synbiotic microcapsule concentration needed to reach 50% inhibition of the initial enzyme reaction rate.

### 2.5. In Vitro Simulation of Gastrointestinal Digestion

The in vitro simulated digestion method by Gomez-Mascaraque, Morfin, Pérez-Masiá, Sanchez, and Lopez-Rubio [31] involved several steps: Simulated Gastric Fluid (SGF) was prepared by mixing pepsin and sodium chloride in deionized water and adjusting its pH to 2.0 using hydrochloric acid. Simulated Intestinal Fluid (SIF) was created by adding pancreatin and porcine bile salt to a phosphate-buffered saline (PBS) solution at pH 7.5. Both SGF and SIF were preheated to 37 °C, filtered, and used for digestion. A powdered sample containing free and encapsulated cells was mixed with SGF for gastric digestion at 37 °C, with samples withdrawn at 30, 60, 90, and 120 min. After gastric digestion, SIF was added, and samples were retrieved at 1, 2, 3, and 4 h for simulated intestinal digestion, followed by cell counting.

(8)
$$\mathrm{Viability}=\frac{log\;cfu\;N1}{log\;cfu\;N0}$$


(9)
Reduction in cell numbers=Survival rate (initial time)−Survival rate (SGF and SIF)
where N1 represents the count of probiotic cells at the time of digestion in simulated gastric fluid (SGF) and simulated intestinal fluid (SIF), and N0 represents the count of probiotic cells at the initial time (0 h).

### 2.6. Statistical Analysis

Data collected in triplicate underwent analysis using a one-way analysis of variance (ANOVA). The detection of significant differences among means (*p* ≤ 0.05) was conducted using the Duncan procedure, employing SPSS 16.0 for Windows (SPSS Inc., Chicago, IL, USA).

## 3. Results and Discussion

### 3.1. The Effects of Microencapsulation and Spray Drying on the Survival Rate

Table 1 presents the survival rates of *B. coagulans* cells after encapsulation using various wall-matrix ratios and subsequent spray drying. Initially, the probiotic count was consistent at approximately 10.35 log CFU/g across all samples. Our investigation revealed a significant decrease in the viability of free cells after the spray-drying process, showing a reduction of about 4.67 log CFU/g. This decline in viability during spray drying can be attributed to factors such as heat-induced stress, desiccation, and exposure to ambient oxygen, all of which collectively impact the metabolic activity of microbial cells, leading to decreased viability.

Interestingly, an increase in the concentration of skim milk powder within the encapsulant agents corresponded to improved viability of the encapsulated probiotics. Specifically, SMC1, containing 23.75% skim milk powder, exhibited a viability rate of 66.73%. Similarly, SMC3 (with 47.50% skim milk powder), SMC5 (containing 59.38% skim milk powder), and SMC7 (with 66.50% skim milk powder) displayed viability rates of 78.70%, 92.37%, and 92.34%, respectively. Consequently, post-spray drying, different wall matrices, designated as SMC1-7, showcased a spectrum of probiotic survival rates.

These findings unmistakably delineate a robust association between the concentration of skim milk powder and the viability rate of the probiotics enclosed within. It is noteworthy that when skim milk powder is employed in limited proportions, it may prove inadequate in constructing the requisite stable and protective encasements around the microbial cells, potentially leading to an encumbered encapsulation process wherein the cells are not fully enveloped or adequately shielded within the capsules [36]. Conversely, elevated concentrations of skim milk powder may furnish a surfeit of both quantity and quality in terms of wall materials, thereby facilitating effective encapsulation, averting cell diffusion, and preserving viability [37].

In addition, proteins can effectively shield probiotic cells from the detrimental effects of high temperatures during the spray-drying process. This thermal protection is crucial for preserving the viability and functionality of probiotics [24]. This study suggests that the choice of encapsulation material can have a significant impact on the viability of probiotic microorganisms. This study emphasizes the impact of encapsulation materials on probiotic viability. *B. coagulans* showed higher survival (>66%) with SMC1-7 encapsulation, surpassing *L. acidophilus* encapsulated with maltodextrin, skim milk powder, and trehalose, which had a 59.2% survival rate, aligning with Soukoulis et al. [38]’s findings. Consequently, this study has brought to light that the conditions characterized by SMC5 and SMC7 emerge as the optimal parameters for microencapsulation, particularly with regard to the augmentation of *B. coagulans* spore viability post-spray-drying operation.

### 3.2. Bioactivity of the Encapsulated Products

#### 3.2.1. In Vitro Antioxidant Activities

Oxidative stress, marked by increased levels of reactive oxygen species (ROS), is correlated with diverse forms of cellular damage and chronic diseases such as cardiovascular diseases in humans, diabetes, neurodegenerative diseases, and cancer [39]. While synthetic antioxidants have traditionally been employed to protect against oxidative stress, concerns have arisen regarding their safety and long-term effects. Consequently, researchers have focused on natural antioxidant solutions, including probiotics [39]. The research conducted by Kodali and Sen [40] has uncovered multiple beneficial properties of *B. coagulans*, including its antibacterial, antiviral, and antioxidant activities. Furthermore, the study demonstrates that supplementing with *B. coagulans* effectively reduces NH_3_ levels in the serum, thereby alleviating oxidative stress [41]. However, numerous chemical analyses employing diverse techniques have been performed to evaluate the antioxidant properties of the compounds of interest. The incorporation of these chemical analyses, which operate via a variety of mechanisms, can be instrumental in shedding light on the primary modes of action of these antioxidant compounds.

In our experimental findings, it was discerned that the encapsulated samples, denoted as SMC1-7, exhibited a notable reduction in the EC_50_ value in comparison to the free cells, as illustrated in Figure 3A–C. These observations underscore the advantageous influence of encapsulation on the antioxidant activity exhibited by *B. coagulans* metabolites and spores, particularly in their proficiency to counteract DPPH radicals, hydroxyl radicals, and superoxide radicals. Intriguingly, both SMC5 and SMC7 manifested the most diminished EC_50_ values for DPPH and hydroxyl radical-scavenging activities when juxtaposed with the other specimens. A lower EC_50_ value signifies a heightened potency in the scavenging activity against hydroxyl radicals, indicative of their pronounced efficacy in the realm of antioxidants, particularly concerning mechanisms involving hydrogen atom transfer (HAT), single electron transfer (SET), and the quenching of hydroxy radicals. Among oxygen radicals, hydroxyl radicals are renowned for their extraordinary reactivity and their proclivity to inflict extensive damage upon adjacent biomolecules. This oxidative damage encompasses processes associated with aging, the genesis of cancer, and the onset of various pathological conditions [42]. Furthermore, SMC5 exhibited the most remarkable superoxide radical-scavenging activity, with the lowest EC_50_ value recorded at 0.82 mg/mL. This reactive oxygen species plays a crucial role in oxidative stress. In our study, we observed that the microencapsulated product of *B. coagulans* TISTR 1447 exhibited higher bioactive properties compared to *B. coagulans* T242, as indicated by the research conducted by Sui et al. [43]. Specifically, within the range of 10^6^–10^8^ CFU/g, *B. coagulans* TISTR 1447 showed superior DPPH-scavenging activity (35.0%), hydroxyl radical-scavenging activity (39.0%), and superoxide anion radical-scavenging activity (14.8%) in comparison to *B. coagulans* T242. In addition, probiotics, such as *B. coagulans*, are renowned for their capacity to generate antioxidant compounds, including butyrate, glutathione (GSH), and folate [44]. These metabolites play a role in diminishing oxidative stress and improving the assimilation of dietary antioxidants via indirect mechanisms [40,45]. Furthermore, the study conducted by Kodali and Sen [40] sheds light on the role of *B. coagulans* in influencing the composition and activity of the gut microbiota. A balanced gut microbiota is crucial for maintaining an overall antioxidant balance, and *B. coagulans* may enhance antioxidant activity by promoting a healthy balance of gut bacteria.

Nonetheless, our observations revealed a noteworthy distinction in the antioxidant potential between the control sample, consisting of encapsulated materials devoid of *B. coagulans* metabolites and spores, and both the free-cell and encapsulated-probiotic-cell counterparts. The control samples containing skim milk powder exhibited markedly diminished antioxidant potency, as evidenced by the highest recorded EC_50_ value in comparison to the free-cell and encapsulated-probiotic-cell variants. This peculiarity in the control samples’ antioxidant activity can be ascribed to the presence of casein compounds within skim milk powder. Skimmed milk can serve as a protective matrix for probiotic bacteria; the probiotic microorganisms are encapsulated or embedded within the milk matrix. This protective matrix helps shield the probiotics from environmental stresses, such as exposure to oxygen, light, or temperature fluctuations, which can lead to oxidative damage [46]. Specifically, the amino acid composition of casein encompasses amino acids endowed with antioxidant properties, such as cysteine and methionine, integrated into its molecular structure [47]. Consequently, our investigation unequivocally elucidates that the SMC5 condition emerges as the preeminent choice for the development of microencapsulated *B. coagulans* products endowed with heightened antioxidative capabilities, equipping them with the capacity to counteract the deleterious effects of free radicals.

#### 3.2.2. α-Amylase and α-Glucosidase Inhibitory Activities

Diabetes is a chronic metabolic disease marked by elevated blood glucose levels, resulting in long-term damage to the heart, blood vessels, eyes, kidneys, and nerves. One primary approach to diabetes management involves inhibiting the digestion and absorption of carbohydrates, such as α-amylase, which gradually increases blood sugar levels after meals. It can effectively control post-meal blood glucose spikes [48]. Its primary function occurs within the small intestine, where it enzymatically breaks down disaccharides and complex carbohydrates, converting them into simpler sugars, such as glucose. The inhibition of α-glucosidase leads to a delay in the absorption of glucose from the gastrointestinal tract into the bloodstream, contributing to the reduction of post-meal blood sugar levels [49].

Within the ambit of our study, it was ascertained that all microencapsulated products exhibited a heightened degree of inhibitory activity against α-amylase in comparison to their free-cell counterparts. Among the various microcapsules investigated, SMC1 displayed the least inhibition of α-amylase activity, even when juxtaposed against all other microcapsules, as graphically represented in Figure 4A. Remarkably, SMC5 showcased the most robust inhibitory activity against α-amylase, as substantiated by an IC_50_ value of 1.32 mg/mL. Intriguingly, these observed activities appeared to remain unaffected by alterations in the ratio of skim milk within the microcapsules. Notably, SMC5′s inhibitory potency against α-amylase surpassed that of acarbose, a pharmaceutical agent commonly employed for the management of diabetes via the inhibition of carbohydrate digestion. The IC_50_ value for acarbose stood at 1.42 mg/mL. Our findings harmonize with those delineated by Li et al. [50], who also documented a favorable correlation between the presence of *B. coagulans* and the inhibition of α-glucosidase activity.

Among the diverse array of microencapsulated products examined, SMC5 distinctly emerges as the frontrunner, showcasing the most pronounced α-glucosidase-inhibitory activity, characterized by an exceptionally low IC_50_ value of 1.15 mg/mL, as graphically delineated in Figure 4B. These observations suggest that microcapsules possess the capacity to retard the absorption of glucose from the intestinal tract into the circulatory system. This phenomenon holds promise for the management of postprandial blood glucose levels. It is of paramount importance to underscore, however, that SMC5, despite its commendable efficacy, still exhibits a lower potential in comparison to acarbose, a synthetic α-glucosidase inhibitor boasting an IC_50_ value of 0.033 mg/mL. Studies have reported similar results regarding the inhibition of α-glucosidase activity in *B. coagulans* CC spores. Additionally, *B. subtilis* B2 from fermented food demonstrated the ability to produce an α-glucosidase inhibitor, as documented by Kim et al. [51] and Zhu et al. [52]. Probiotics, including *B. coagulans,* may produce inhibitory compounds such as organic acids, bacteriocins, short-chain fatty acids (SCFAs), or peptides that can interfere with the activities of α-amylase and α-glucosidase. These compounds may directly bind to the enzymes, rendering them less active [53]. Consequently, our research underscores the paramount suitability of SMC5 for the development of microencapsulated *B. coagulans* products with antidiabetic attributes, endowing them with the capacity to inhibit both α-amylase and α-glucosidase activities.

#### 3.2.3. DPP-IV-Inhibitory Activity

DPP-IV is a crucial enzyme involved in the breakdown of incretin hormones like glucagon-like peptide-1 (GLP-1), which plays a key role in regulating blood sugar levels. The inhibition of DPP-IV results in elevated levels of active GLP-1, which, in turn, promotes increased insulin secretion and a reduction in blood sugar levels [54].

In our empirical findings, it was discerned that among the gamut of microcapsules under scrutiny, SMC7 notably exhibited the highest degree of DPP-IV-inhibitory activity, as underscored by its IC_50_ value of 0.25 mg/mL (Figure 5). Subsequently, SMC5 emerged as the second-most efficacious, yielding an IC_50_ value of 0.37 mg/mL. When assessing the overarching proficiency in enzyme inhibition across diverse wall matrices, the hierarchy of efficacy was delineated as follows: SMC7 > SMC5 > SMC3 > SMC1 > free cells > control samples (comprising encapsulated materials bereft of *B. coagulans* spores). This study found a direct improvement in the DPP-4 inhibition of the encapsulated products with an increasing ratio of skim milk. This suggests that skim milk might safeguard the bioactive material responsible for DPP-IV-inhibitory activity. Interestingly, this trend was not observed in the control samples, indicating a unique effect specific to the encapsulated products. However, the control samples exhibited some degree of inhibitory activity, which was attributed to the presence of casein components in skim milk powder. These casein components contain small peptides that have been extensively studied for their various bioactive properties, including potential health benefits such as reducing hypertension (high blood pressure) and modulating the immune system [55]. Our study’s findings align with previous research by Wu et al. [56], who reported similar results for *B. amyloliquefaciens* demonstrating DPP-4-inhibitory activity. Moreover, probiotic bacteria have the ability to produce a range of bioactive compounds, such as peptides and metabolites, which are capable of inhibiting DPP-4 activity. These compounds can directly interact with the enzyme, diminishing its capacity to break down incretin hormones like GLP-1, as demonstrated by Yan et al. [10]. This interaction highlights the potential significance of probiotic compounds in regulating DPP-4 activity and preserving incretin hormones. Our findings align with the work conducted by Mudgil et al. [57], who reported the plausible DPP-IV-inhibitory activity of probiotic strains, suggesting potential improvements in blood glucose regulation. These insights hold implications for ameliorating fasting blood glucose levels in individuals afflicted with type 2 diabetes and enhancing glucose tolerance. Such discernments are poised to inform dietary choices for diabetic patients and contribute to the development of safer and more efficacious pharmaceutical agents in the foreseeable future.

The encapsulation processes of SMCs played a vital role in improving their biological activities. These bioactivities were found to correlate with survival rates, as depicted in Table 1. Notably, SMC5 and SMC7 exhibited the highest survival rates after the spray-drying process. This observation suggests that skim milk is a critical component for their effectiveness and demonstrates stronger inhibitory activity. As a result, these two conditions hold the potential to offer enhanced therapeutic benefits, especially in processes like the spray drying of *B. coagulans*.

### 3.3. The Survival Rate during In Vitro Simulation of Gastrointestinal Digestion

Figure 6 portrays the viability profiles of both free and microencapsulated *B. coagulans* cells, employing various wall materials denoted as SMCs, throughout a six-hour simulation mimicking gastrointestinal digestion. The free cells demonstrated a decline in viability amounting to 3.2 log CFU/g over the entire duration of the digestion process. Of particular note, all samples encapsulated within SMCs exhibited markedly superior survival rates subsequent to the simulated gastrointestinal digestion in comparison to their free-cell counterparts. This enhancement was substantiated by a diminution in viability of less than 1.0 log CFU/g. These findings underscore the efficacy of microencapsulation employing a protein-based coating augmented with prebiotic inulin in significantly bolstering the survivability of probiotic entities within the simulated gastric and intestinal environments, surpassing the resilience of unencapsulated spores.

Notably, these rates still exceeded the recommended 6 log CFU/g set by the International Dairy Federation (Table 1). The SMC7 sample exhibited the highest cell survival, with only a decrease of 0.58 log CFU/g during gastrointestinal digestion, followed by the SMC5 sample, which had a nonsignificant difference from SMC7 with a loss of 0.93 log CFU/g, and the SMC1 sample, with a loss of 1.14 log CFU/g. Our study yielded more robust results compared to those presented by Yoha et al. [58]. They found that encapsulating *L. plantarum* using a combination of fructo-oligosaccharide, whey protein, and maltodextrin led to a significant loss of cell viability, with reductions of 2–3 log during exposure to gastric conditions and 4 log under small-intestinal conditions. In contrast, our research demonstrated that skim milk proteins and cellulose acetate create a protective barrier around probiotics. This barrier potentially reduces their exposure to proteolytic enzymes in the digestive system, enhancing the probiotics’ survival and enabling them to maintain their activity within the gut [59].

Our experiments have conclusively confirmed that microencapsulation acts as a vital shield for the *B. coagulans* probiotic, safeguarding it from the harsh gastric environment characterized by low pH levels and the presence of digestive enzymes in the intestinal tract. To achieve this, we utilized cellulose acetate phthalate, a well-known polymer recognized for its ability to control pH-dependent releases. This material dissolves under mildly acidic or neutral intestinal conditions yet remarkably withstands highly acidic gastric fluids [26]. This exceptional quality has the potential to significantly slow the release of *B. coagulans* spores and metabolite materials during the gastric phase, ensuring a more controlled release in the intestinal phase. This protective mechanism not only guarantees the probiotic’s efficacy but also holds the promise of delivering numerous health benefits upon oral consumption. Importantly, our findings shed light on the intricate effects of microencapsulation, revealing its dependence on the careful selection of coating materials and the specific physiological conditions encountered within the digestive tract.

Furthermore, our investigation sought to evaluate the bioactive activity of these encapsulated samples during GI digestion; however, it was observed that these samples contained a high concentration of gastrointestinal salts (sodium chloride and phosphate-buffered saline), gastrointestinal enzymes (pepsin and pancreatin), and bile salts (porcine bile salt). Consequently, our research focused exclusively on the assessment of survival rates, given the substantial interference posed by the factors mentioned above on the bioactive potential of the probiotics.

## 4. Conclusions

The optimal ratio for successful spray drying of *B. coagulans* was developed, resulting in the creation of a stable encapsulated synbiotic powder with improved survival rates and enhanced bioactive properties. The proportion of skim milk powder in the formulation was increased, proving beneficial as it bolstered the survival of *B. coagulans* spores during the spray-drying process, enabling them to withstand the rigors of gastrointestinal digestion. Particularly noteworthy was the robust survival of *B. coagulans* when exposed to simulated intestinal fluid, demonstrating its potential in the gut. Furthermore, heightened biological activities were observed in the encapsulated products. These included effective antioxidant activity against substances like DPPH, hydroxy radicals, and superoxide radicals, as well as noteworthy antidiabetic effects, including the inhibition of α-amylase and α-glucosidase enzymes. Among the various conditions tested, SMC5 exhibited the most impressive performance in terms of protecting *B. coagulans* cells and delivering multifunctional bioactive properties. The research-derived formula has the potential to benefit a wide range of probiotics, suggesting exciting prospects for the development of commercial products via further exploration and optimization.

## Figures and Tables

**Figure 1 foods-12-03692-f001:**
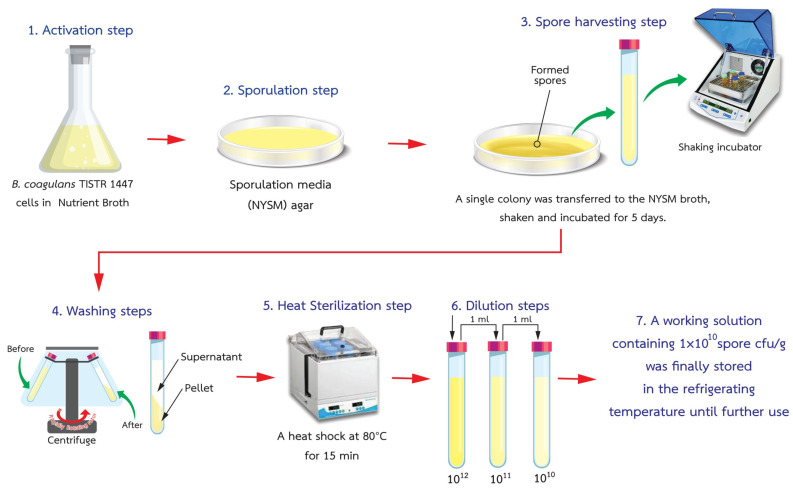
Diagram illustrating *B. coagulans* spore production.

**Figure 2 foods-12-03692-f002:**
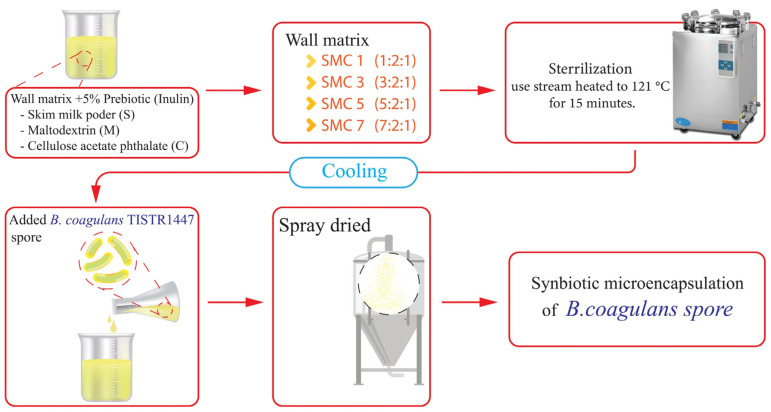
Diagram depicting synbiotic microcapsules produced via a spray-drying process.

**Figure 3 foods-12-03692-f003:**
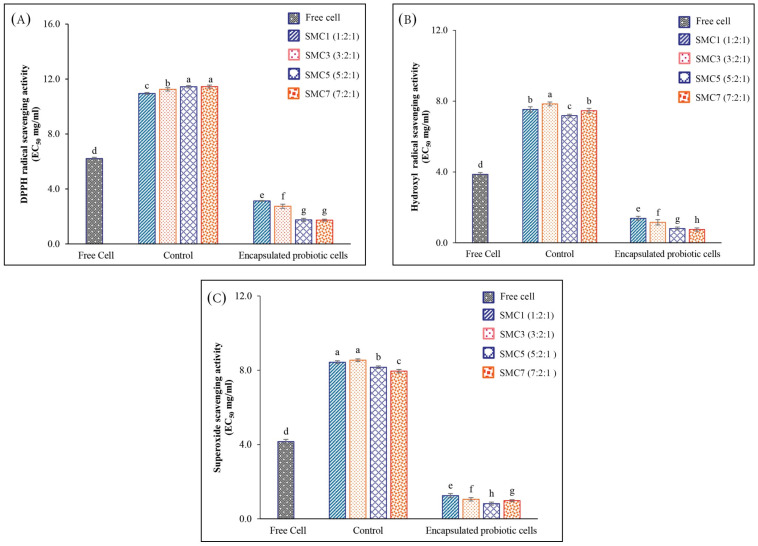
The graph illustrates the in vitro antioxidant activity of both free and encapsulated probiotic cells within different wall matrices, assessed using distinct methods: DPPH (**A**), hydroxyl radicals (**B**), and superoxide anion radicals (**C**). Values with notable differences (*p* ≤ 0.05) are indicated by individual superscript letters (a–h).

**Figure 4 foods-12-03692-f004:**
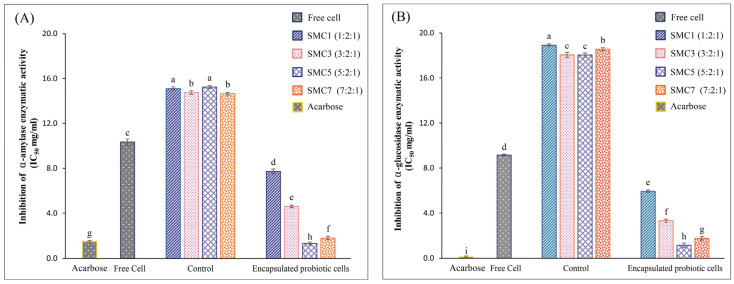
In vitro antidiabetic activity of free and encapsulated probiotic cells in different wall matrices via α-amylase (**A**) and α-glucosidase (**B**). Values with notable differences (*p* ≤ 0.05) are indicated by individual superscript letters (a–i).

**Figure 5 foods-12-03692-f005:**
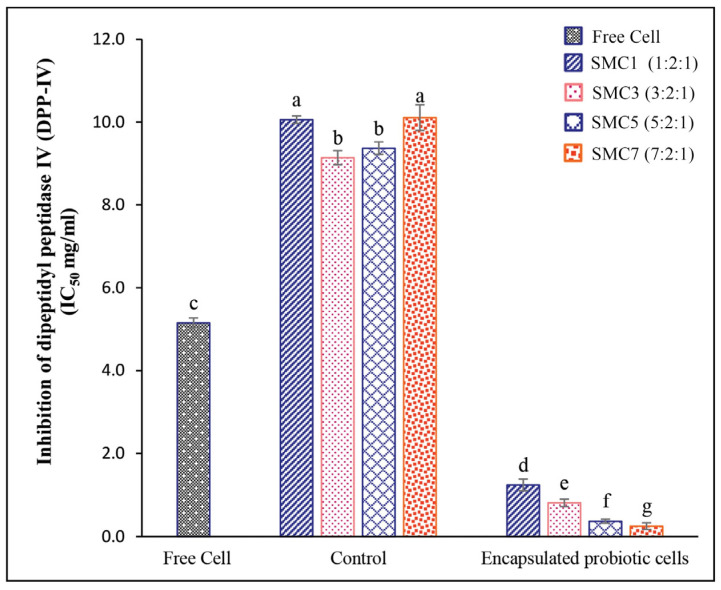
In vitro DPP-IV-inhibitory activity of free and encapsulated probiotic cells in different wall matrices. Values with notable differences (*p* ≤ 0.05) are indicated by individual superscript letters (a–g).

**Figure 6 foods-12-03692-f006:**
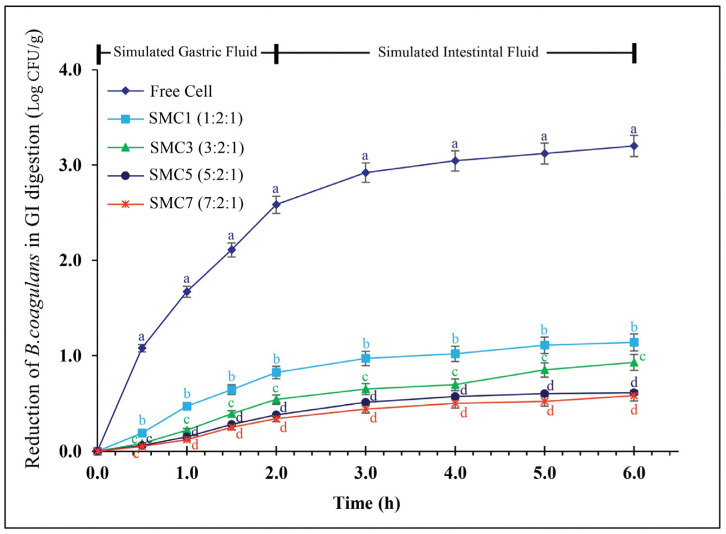
The reduction in *B. coagulans* during *in vitro*-simulated gastrointestinal digestion of free and encapsulated cells with various wall materials. Values with notable differences (*p* ≤ 0.05) are indicated by individual superscript letters (a–d) at the same time point (h).

**Table 1 foods-12-03692-t001:** The survival rates of free cells and cells encapsulated in various wall-matrix ratios before and after the spray-drying process.

Samples	Viable Cells (log CFU/g)		Survival Rate (%)
	Before Spray Drying	After Spray Drying	
Free Cell	10.35 ± 0.15 ^a^	4.67 ± 0.26 ^d^	45.12 ± 0.25 ^d^
SMC1 (1:2:1)	10.91 ± 0.29 ^a^	7.28 ± 0.37 ^c^	66.73 ± 0.73 ^c^
SMC3 (3:2:1)	10.61 ± 0.93 ^a^	8.35 ± 0.20 ^b^	78.70 ± 0.18 ^b^
SMC5 (5:2:1)	10.75 ± 0.06 ^a^	9.93 ± 0.53 ^a^	92.37 ± 0.36 ^a^
SMC7 (7:2:1)	10.58 ± 0.11 ^a^	9.77 ± 0.94 ^a^	92.34 ± 0.27 ^a^

Significant differences within the column (*p* ≤ 0.05) are denoted by superscript letters (a–d).

## Data Availability

The article contains the data and materials supporting the conclusions of this study.

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
