# Peer review of "Innovative Insights for Establishing a Synbiotic Relationship with Bacillus coagulans: Viability, Bioactivity, and In Vitro-Simulated Gastrointestinal Digestion"

_foods, 2023, doi:10.3390/foods12193692_

Round 1

Reviewer 1 Report

 This work explored different combinations of encapsulating materials via spray-drying for improving survival of probiotic B. coagulans . The present study is very interesting and well-written but some issues should be addressed/corrected:

Line 41 – The mention Poshadri, et al. should be deleted.

Line 85 - The bacterial species Lactobacillus plantarum should be written in italic.

Lines 382- 391 – The information reported in these lines, citing “Among the various microcapsulated products, SMC5 stands out for displaying the highest α-glucosidase inhibitory activity, with a notably low IC50 value of 1.15 mg/ml, as demonstrated in Figure 3B. These imply that microcapsules have the potential to delay the absorption of glucose from the intestines into the bloodstream, which could be advantageous in managing post-meal blood sugar levels. It's important to highlight, however, that SMC5 still exhibits lower efficacy compared to acarbose, a synthetic α-glucosidase inhibitor with an IC50 value of 0.033 mg/ml (data not presented). Consequently, our research underscores SMC5 as the preferred choice for developing microencapsulated B. coagulans products with antidiabetic properties, enabling them to inhibit both α-amylase and α-glucosidase activities proficiently.” was repeated in the next paragraph, citing “Among the diverse array of microencapsulated products examined, SMC5 distinctly emerges as the frontrunner, showcasing the most pronounced α-glucosidase inhibitory activity, characterized by an exceptionally low IC50 value of 1.15 mg/ml, as graphically delineated in Figure 3B. These observations suggest that microcapsules possess the capacity to retard the absorption of glucose from the intestinal tract into the circulatory system. This phenomenon holds promise in the management of postprandial blood glucose levels. It is of paramount importance to underscore, however, that SMC5, despite its commendable efficacy, still exhibits a lower potential in comparison to acarbose, a synthetic α-glucosidase inhibitor boasting an IC50 value of 0.033 mg/ml (data not presented herein). Consequently, our research underscores the paramount suitability of SMC5 for the development of microencapsulated B. coagulans products with antidiabetic attributes, endowing them with the capacity to inhibit both α-amylase and α-glucosidase activities.” The authors should be readapted these two paragraphs in order to avoid repetition of information.

Lines 423 -425 – The following sentence is confusing, “It was observed that the concentration of skim milk bore a dose-dependent influence on the enhancement of DPP-IV inhibition. “ because it is not observe a trend/correlation between DPP-IV inhibition and skim milk concentration in encapsulating matrix. The authors should clarify.

Lines 425-429 The following sentence is unclear, and the authors should clarify “Additionally, the control samples exhibited some degree of inhibitory activity, attributed to the presence of casein components replete with small peptides. These peptides, coexistent with casein, have been extensively investigated for their multifaceted bioactive attributes, encompassing potential health benefits such as hypertension mitigation and immune system modulation [42]. “

Lines 443-444 – The sentence “The encapsulation processes of SMCs played a vital role in improving their bioactive activities.” Should be restructured in order to avoid certain repetitions namely bioactive activities should be replaced by biological activities.

Line 453- The authors should be uniformize the term free cells throughout the manuscript, avoiding synonyms such as “unfettered cells”  .

Line 496-497– The authors should be replaced “bioactive activities” by biological activities

Only some corrections of English are needed. 

Author Response

Dear Reviewer,

I am submitting a revised manuscript titled "Innovative Insights for Establishing a Synbiotic Relationship with Bacillus coagulans: Viability, Bioactivity, and In Vitro-Simulated Gastrointestinal Digestion" for consideration in Foods.

The manuscript has been extensively revised per the suggestions/comments of reviewers. The changes were highlighted in the color front. Point-by-point responses are also submitted in a separate file to the Reviewer. Additional results requested by the Reviewer are also presented in the manuscript. I sincerely thank each Reviewer for your time and efforts on this manuscript. We hope that our revised manuscript will meet the journal's requirements.

Our study explores the potential of synbiotic microcapsules, emphasizing the use of encapsulating agents to establish a strong association with Bacillus coagulans, a probiotic of great significance in functional foods. We investigated various encapsulation technologies, assessing materials like skim milk powder, maltodextrin, and cellulose acetate phthalate (SMC1, SMC3, SMC5, and SMC7). Our formulations, incorporating 5% inulin as a prebiotic, demonstrated exceptional preservation of B. coagulans spores during spray drying, with formulations SMC5 and SMC7 displaying the highest viability. SMC5 exhibited remarkable antioxidant and antidiabetic properties. This study not only impacts probiotics and functional foods but also extends to food science and nutrition. Synbiotic microcapsules containing SMC5 offer advanced protection and maintain bioactivity, marking a significant advancement in functional food development. Our comprehensive approach integrating encapsulation, probiotics, and prebiotics enriches understanding in these areas. We believe our findings align with the objectives of Foods, contributing substantially to its academic and practical relevance.

Thank you for your consideration.

Yours sincerely,

Papungkorn Sangsawad

Point 1: Line 41 – The mention of Poshadri, et al. should be deleted.

Response 1: The mention of Poshadri, et al. was deleted (line 42).

Point 2: Line 85 - The bacterial species Lactobacillus plantarum should be written in italic.

Response 2: The Lactobacillus plantarum was italicized (line 93).

Point 3: Lines 382- 391 – The information reported in these lines, citing “Among the various microcapsulated products, SMC5 stands out for displaying the highest α-glucosidase inhibitory activity, with a notably low IC50 value of 1.15 mg/ml, as demonstrated in Figure 3B. These imply that microcapsules have the potential to delay the absorption of glucose from the intestines into the bloodstream, which could be advantageous in managing post-meal blood sugar levels. It's important to highlight, however, that SMC5 still exhibits lower efficacy compared to acarbose, a synthetic α-glucosidase inhibitor with an IC50 value of 0.033 mg/ml (data not presented). Consequently, our research underscores SMC5 as the preferred choice for developing microencapsulated B. coagulans products with antidiabetic properties, enabling them to inhibit both α-amylase and α-glucosidase activities proficiently.” was repeated in the next paragraph, citing “Among the diverse array of microencapsulated products examined, SMC5 distinctly emerges as the frontrunner, showcasing the most pronounced α-glucosidase inhibitory activity, characterized by an exceptionally low IC50 value of 1.15 mg/ml, as graphically delineated in Figure 3B. These observations suggest that microcapsules possess the capacity to retard the absorption of glucose from the intestinal tract into the circulatory system. This phenomenon holds promise in the management of postprandial blood glucose levels. It is of paramount importance to underscore, however, that SMC5, despite its commendable efficacy, still exhibits a lower potential in comparison to acarbose, a synthetic α-glucosidase inhibitor boasting an IC50 value of 0.033 mg/ml (data not presented herein). Consequently, our research underscores the paramount suitability of SMC5 for the development of microencapsulated B. coagulans products with antidiabetic attributes, endowing them with the capacity to inhibit both α-amylase and α-glucosidase activities.” The authors should be readapted these two paragraphs in order to avoid repetition of information.

Response 3: Lines 438-456 were deleted and replaced by this paragraph: " Among the diverse array of microencapsulated products examined, SMC5 distinctly emerges as the frontrunner, showcasing the most pronounced α-glucosidase inhibitory activity, characterized by an exceptionally low IC50 value of 1.15 mg/ml, as graphically de-lineated in Figure 4B. These observations suggest that microcapsules possess the capacity to retard the absorption of glucose from the intestinal tract into the circulatory system. This phenomenon holds promise in the management of postprandial blood glucose levels. It is of paramount importance to underscore, however, that SMC5, despite its com-mendable efficacy, still exhibits a lower potential in comparison to acarbose, a synthetic α-glucosidase inhibitor boasting an IC50 value of 0.033 mg/ml. Studies have reported similar results regarding the inhibition of α-glucosidase activity in B. coagulans CC spores. Additionally, B. subtilis B2 from fermented food demonstrated the ability to produce an α-glucosidase inhibitor, as documented by Kim, et al. [50] and Zhu, et al. [51]. Probiotics, including B. coagulans, may produce inhibitory compounds such as organic acids, bacteriocins, short-chain fatty acids (SCFAs), or peptides that can interfere with the activity of α-amylase and α-glucosidase. These compounds may directly bind to the enzymes, rendering them less active [52]. Consequently, our research underscores the paramount suit-ability of SMC5 for the development of microencapsulated B. coagulans products with antidiabetic attributes, endowing them with the capacity to inhibit both α-amylase and α-glucosidase activities."

Point 4: Lines 423 -425 – The following sentence is confusing, “It was observed that the concentration of skim milk bore a dose-dependent influence on the enhancement of DPP-IV inhibition. “ because it is not observe a trend/correlation between DPP-IV inhibition and skim milk concentration in encapsulating matrix. The authors should clarify.

Response 4:  The paragraph was revised and clarified (lines 473-482).

Point 5: Lines 425-429 -The following sentence is unclear, and the authors should clarify “In our study, we observed that the microencapsulated product of B. coagulans TISTR 1447 exhibited higher bioactive properties compared to B. coagulans T242, as indicated by the research conducted by Sui, et al. [1]. “

Response 5:  The paragraph was revised and clarified (lines 377-380).

Point 6: Lines 443-444 -The sentence “The encapsulation processes of SMCs played a vital role in improving their bioactive activities.” Should be restructured in order to avoid certain repetitions namely bioactive activities should be replaced by biological activities.

Response 6:  biological activities were replaced (lines 500).

Point 7: Line 453- The authors should be uniformize the term free cells throughout the manuscript, avoiding synonyms such as “unfettered cells”  .

Response 7:  The unfettered cells were replaced with free cells (lines 511).

Point 8: Line 496-497– The authors should be replaced “bioactive activities” by biological activities

Response 8:  The bioactive activities were replaced biological activities (lines 501).

Reviewer 2 Report

The authors wrote high quality manuscript, with very interesting results and significant content. Some minor remark should be addressed. I suggest authors to improve the comparison of the obtained results with literature data.

Author Response

Dear Reviewer,

I am submitting a revised manuscript titled "Innovative Insights for Establishing a Synbiotic Relationship with Bacillus coagulans: Viability, Bioactivity, and In Vitro-Simulated Gastrointestinal Digestion" for consideration in Foods.

The manuscript has been extensively revised per the suggestions/comments of reviewers. The changes were highlighted in the color front. Point-by-point responses are also submitted in a separate file to the Reviewer. Additional results requested by the Reviewer are also presented in the manuscript. I sincerely thank each Reviewer for your time and efforts on this manuscript. We hope that our revised manuscript will meet the journal's requirements.

Our study explores the potential of synbiotic microcapsules, emphasizing the use of encapsulating agents to establish a strong association with Bacillus coagulans, a probiotic of great significance in functional foods. We investigated various encapsulation technologies, assessing materials like skim milk powder, maltodextrin, and cellulose acetate phthalate (SMC1, SMC3, SMC5, and SMC7). Our formulations, incorporating 5% inulin as a prebiotic, demonstrated exceptional preservation of B. coagulans spores during spray drying, with formulations SMC5 and SMC7 displaying the highest viability. SMC5 exhibited remarkable antioxidant and antidiabetic properties. This study not only impacts probiotics and functional foods but also extends to food science and nutrition. Synbiotic microcapsules containing SMC5 offer advanced protection and maintain bioactivity, marking a significant advancement in functional food development. Our comprehensive approach integrating encapsulation, probiotics, and prebiotics enriches understanding in these areas. We believe our findings align with the objectives of Foods, contributing substantially to its academic and practical relevance.

Thank you for your consideration.

Yours sincerely,

Papungkorn Sangsawad

Point 1: I suggest authors to improve the comparison of the obtained results with literature data.

Response 1: 

3.1 The effects of microencapsulation and spray drying on the survival rate

              The study suggests that the choice of encapsulation material can have a significant impact on the viability of probiotic microorganisms. The study emphasizes the impact of encapsulation materials on probiotic viability. B. coagulans showed higher survival (>66%) with SMC1-7 encapsulation, surpassing L. acidophilus encapsulated with maltodextrin, skim milk powder, and trehalose, which had a 59.2% survival rate, aligning with Soukoulis, et al. [2] findings. (line336-341)

3.2.1 In vitro antioxidant activities

In our study, we observed that the microencapsulated product of B. coagulans TISTR 1447 exhibited higher bioactive properties compared to B. coagulans T242, as indicated by the research conducted by Sui, Zhu, Wu, Ma, Tuo, Jiang, Qian and Mu [1]. Specifically, within the range of 106-108 CFU/g, B. coagulans TISTR 1447 showed superior DPPH scavenging activity (35.0%), hydroxyl radical scavenging activity (39.0%), and superoxide anion radical scavenging activity (14.8%) in comparison to B. coagulans T242. (line 377-383).

3.2.2 α-amylase and α-glucosidase inhibitory activities

Studies have reported similar results regarding the inhibition of α-glucosidase activity in B. coagulans CC spores. Additionally, B. subtilis B2 from fermented food demonstrated the ability to produce an α-glucosidase inhibitor, as documented by Kim, et al. [3] and Zhu, et al. [4]. (line 446-449)

3.2.3 DPP-IV Inhibitory activity

Our study's findings align with previous research by Wu, et al. [5], which reported similar results for B. amyloliquefaciens, demonstrating DPP-4 inhibitory activity. (line482-484)

3.3 The survival rate during in vitro simulation of gastrointestinal digestion

Our study yielded more robust results compared to those presented by Yoha, et al. [6]. They found that encapsulating L. plantarum using a combination of fructooligosaccharide, whey protein, and maltodextrin led to a significant loss of cell viability, with reductions of 2-3 log during exposure to gastric conditions and 4 log in small intestinal conditions. In contrast, our research demonstrated that skimmed milk proteins and cellulose acetate could create a protective barrier around probiotics. This barrier potentially reduces their exposure to proteolytic enzymes in the digestive system, enhancing probiotics' survival and enabling them to maintain their activity within the gut [7].(line524-532)

Reviewer 3 Report

In the attached file.

The English is sound and conveys all information well. There were very few issues with the style. I have mentioned them in the specific comments.

Author Response

Dear Reviewer,

I am submitting a revised manuscript titled "Innovative Insights for Establishing a Synbiotic Relationship with Bacillus coagulans: Viability, Bioactivity, and In Vitro-Simulated Gastrointestinal Digestion" for consideration in Foods.

The manuscript has been extensively revised per the suggestions/comments of reviewers. The changes were highlighted in the color front. Point-by-point responses are also submitted in a separate file to the Reviewer. Additional results requested by the Reviewer are also presented in the manuscript. I sincerely thank each Reviewer for your time and efforts on this manuscript. We hope that our revised manuscript will meet the journal's requirements.

Our study explores the potential of synbiotic microcapsules, emphasizing the use of encapsulating agents to establish a strong association with Bacillus coagulans, a probiotic of great significance in functional foods. We investigated various encapsulation technologies, assessing materials like skim milk powder, maltodextrin, and cellulose acetate phthalate (SMC1, SMC3, SMC5, and SMC7). Our formulations, incorporating 5% inulin as a prebiotic, demonstrated exceptional preservation of B. coagulans spores during spray drying, with formulations SMC5 and SMC7 displaying the highest viability. SMC5 exhibited remarkable antioxidant and antidiabetic properties. This study not only impacts probiotics and functional foods but also extends to food science and nutrition. Synbiotic microcapsules containing SMC5 offer advanced protection and maintain bioactivity, marking a significant advancement in functional food development. Our comprehensive approach integrating encapsulation, probiotics, and prebiotics enriches understanding in these areas. We believe our findings align with the objectives of Foods, contributing substantially to its academic and practical relevance.

Thank you for your consideration.

Yours sincerely,

Papungkorn Sangsawad

Point 1: Title- Instead of placing keywords after : you could consider starting the title by giving precise

information about your encapsulating agent (what is this novel ingredient?).

Response 1:  The authors have changed the title to “Innovative Insights for Establishing a Synbiotic Relationship with Bacillus coagulans: Viability, Bioactivity, and In Vitro-Simulated Gastrointestinal Digestion" (lines 2-4).

Point 2: Abstract- Is slightly chaotic. After you have introduced your formulations, it would be better to include a brief introduction of what you were doing instead of presenting your results straight away. Then you could consider writing only about your choice of the best formulation and rationale behind this choice. The last sentence does not summarise the overall impact of your findings. Perhaps you could consider adding a few words on its novelty in terms of using these specific substances.

Response 2:  The abstract has been revised (lines 18-34). “This study investigates the use of encapsulating agents to establish a synbiotic relationship with Bacillus coagulans. Various ratios of wall materials, such as skim milk powder, maltodextrin, and cellulose acetate phthalate (represented as SMC1, SMC3, SMC5, and SMC7), were examined. In all formulations, 5% inulin was included as a prebiotic. The research assessed their impact on cell viability and bioactive properties during both the spray drying process and in-vitro gastrointestinal digestion. The results demonstrate that these encapsulating agents efficiently protect B. coagulans spores during the spray drying process, resulting in spore viability exceeding 6 log CFU/g. Notably, SMC5 and SMC7 displayed the highest spore viability values. Moreover, SMC5 showcased the most notable antioxidant activity, encompassing DPPH, hydroxy radicals, and superoxide radicals scavenging, as well as significant antidiabetic effects through the inhibition of α-amylase and α-glucosidase. Furthermore, during the simulated gastrointestinal digestion, both SMC5 and SMC7 exhibited a slight reduction in spore viability over the 6-hour simulation. Consequently, SMC5 was identified as the optimal condition for synbiotic production, offering protection to B. coagulans spores during microencapsulation and gastrointestinal digestion while maintaining bioactive properties post-encapsulation. Synbiotic microcapsules containing SMC5 showcased a remarkable positive impact, suggesting its potential as an advanced food delivery system and a functional ingredient for various food products.”

Point 3: Introduction

        - Does not explain why you have chosen these specific encapsulation materials.

This is quite interesting and should be elaborated on.

         Response 3.1:   The authors appreciate your comment. The paragraph has been made more complete, as shown in lines 95-109

  • Sentence in lines 56-57- please, provide a reference

Response 3.2:   The sentence has been referenced (lines 59). 

  • Paragraph in lines 66-81- this is the second time you mention probiotic activities and

microencapsulation in the introduction and there are some style issues here. Please consider

restructuring the introduction so related topics are in the same place

Response 3.3:   The paragraph was revised (lines 59).

  • Sentence in lines 79-81- encapsulation may isolate probiotics from food, but not necessarily. This depends on the coating material, encapsulation technology (if you simply spray dry not all probiotics will be inside of the capsule), storage conditions etc. Please either explain in a more specific manner or remove this claim.

Response 3.4:  The sentence was revised and clarified (lines 82-87).

  • Sentence in lines 94-97- could be stylistically improved

Response 3.4:  The authors have revised and clarified the sentence in lines 103-108. "Researchers have explored multiple wall matrix systems, including ternary wall matrices, to enhance probiotics' viability and physicochemical characteristics [8,9]. Nonetheless, there is a notable gap in the literature regarding the specific combination of encapsulation materials, such as skim milk powder, maltodextrin, and cellulose acetate phthalate, for safeguarding B. coagulans spores during the intricate spray drying process."

Point 4: Materials and methods

Section 2.1 line 112- please mention and reference specific quality standards you may have in mind.

Response 4.1: The sentence was revised to “In this research project, we utilized chemicals of analytical reagent-grade quality.” (lines 125-126)

   Section 2.2- line 118, please explain the abbreviation.

      Response 4.2: The abbreviation has been added (lines 132). “Nutrient Yeast Extract Salt Medium (NYSM) agar.”

               Overall, this section is slightly unclear. What is your final working solution? Why did you subculture your spore suspension on NYESMA (line 125)?

Response 4.3: We apologize for incorrect information. We have removed those messages and revised the information to be more accurate. “Ultimately, the original stock solution underwent dilution, decreasing its concentration from 1×1012 CFU/g to a final concentration of 1×1010 CFU/g. This thinned solution was subsequently stored under refrigeration, prepared for future use, as depicted in Figure 1.” This information has been displayed in lines 139-142

              In my opinion, if you have included a diagram or a figure illustrating the process flow for this procedure it would be much clearer to the reader.

   Response 4.4: The authors appreciate your comment. We took your suggestions into account by adding a diagram, as shown in Figure 1. (line 144)

Figure 1. Diagram illustrating B. coagulans spore production

Section 2.3- Lines 133-138- read as if you have autoclaved your solution together with B.coagulans spores. Please correct, or/and consider adding a flow diagram/ figure illustrating what you have done. Another option is to describe your process in points.

Response 4.5: Section 2.3 was added a flow diagram shown in Figure 2. (line179)

Figure 2. Diagram depicting synbiotic microcapsules produced via spray drying process.

Lines 141 and 151-are the values/ equation variable N0 you are referring to describing the concentration of the microorganisms in the suspension ready for spray-drying or is this calculated on a dry-matter basis? Please explain and if you did not use dry matter basis, then indicate how you corrected for the concentration of probiotics during spray-drying.

Response 4.6: The equation value was determined by evaluating the probiotic cell count (log CFU/g of dry solid material) both prior to and following the spray-drying process. In this context, N(F) denotes the count of probiotic cells after the spray-drying process, while N(0) signifies the count of probiotic cells before the spray-drying procedure. It's noteworthy that this study adheres to the methodology detailed by Gomez-Mascaraque et al. (2016). (lines 170-173).

Lines 152-154- please describe the method used for probiotic enumeration either in a separate section or here.

Response 4.7:  The experiment followed the methodology outlined by Gomez-Mascaraque et al. (2016). The in vitro simulated digestion method by Gomez-Mascaraque et al. (2016). involved several steps: Simulated Gastric Fluid (SGF) was prepared by mixing pepsin and sodium chloride in deionized water, adjusting its pH to 2.0 with hydrochloric acid. Simulated Intestinal Fluid (SIF) was created by adding pancreatin and porcine bile salt to a phosphate-buffered saline (PBS) solution at pH 7.5. Both SGF and SIF were preheated to 37°C, filtered, and used for digestion. A powdered sample containing free and encapsulated cells was mixed with SGF for gastric digestion at 37°C, with samples withdrawn at 30, 60, 90, and 120 minutes. After gastric digestion, SIF was added, and samples were retrieved at 1, 2, 3, and 4 hours for simulated intestinal digestion, followed by cell counting.

                             Viability rate (%) =                                                       (8)

Decreased cell numbers = Survival rate (initial time)- Survival rate (SGF and SIF)              (8)

Where N1 represents the count of probiotic cells at time digestion in simulated gastric fluid (SGF) and simulated intestinal fluid (SIF), and No represents the count of probiotic cells at the initial time (0 h).

 Section 2.4 -what was your sample? You say you have used a certain volume for each test in ul but you have started off with dry capsules, right? How did the sample become liquid?

Response 4.8:  We conducted an analysis of the spray-dried samples, comprising free cells, unencapsulated cells, and encapsulated cells, to assess their bioactive properties. (lines 181-182).

Section 2.5- please add information about the pH in the intestinal digestion phase and how the probiotics were counted.

     Response 4.9:  The sentence was revised and clarified (lines 285, 284, and 292-297).

Section 2.6 sentence in lines 263-264  should be in the past tense, p-value should be ≤ in relation to your significance level.

Response 4.10:  The sentence was revised (line 299-302). “Data collected in triplicate underwent analysis through a one-way analysis of variance (ANOVA). The detection of significant differences among means (p 0.05) was conducted using the Duncan procedure, employing SPSS 16.0 for Windows (SPSS Inc., Chicago, IL, USA).”

Point 5: Results and discussion

Lines 268-277- please reorder text so it reads better (e.g. start with the count in the initial suspension and then describe the reduction rates in each of your treatments.

Response 5.1:  The sentence was revised (line 3076-320).

Lines 279-295- I generally agree with your discussion, but perhaps the nature of the material could also have something to do with its protective effect. You have mentioned the role of protein in the introduction. I think that the references you have mentioned there could be used in this section as well.

Response 5.1:   The paragraph has added the role of protein (lines 333-335) 

Lines 304-313- If B. coagulans was shown to have antioxidative effects in any specific works, then, please state this and support with citations. If not, then state this as well, since your findings may be even more novel.

Response 5.2:   The paragraph was added the antioxidative effects of B. coagulans (lines 351-355). 

Section 3.2.1- overall you report interesting findings but scarcely discuss them. Please include references where the encapsulation material was shown to effectively improve probiotic antioxidative properties. What could be the mechanism behind skimmed milk improving the antioxidative effect of B. coagulans?

Response 5.3: The information provided in Taylor and Richardson [10] study indicated that heat treatment of skim milk increased its antioxidant activity and the presence of 'reactive' sulfhydryl groups. These groups were found to be responsible for the antioxidant activity of skim milk. However, the specific process through which skim milk enhances the antioxidative effect of B. coagulans remains unexplored.

Ref.

Taylor, M.J.; Richardson, T. Antioxidant Activity of Skim Milk: Effect of Heat and Resultant Sulfhydryl Groups. Journal of Dairy Science 1980, 63, 1783-1795,

Lines 369-382- I think that you should show data for acarbose on the graphs. This is your control sample and it is very difficult to compare its result with others shown on the graphs, because of the scale. It would also be appropriate to do statistics showing that there is a significant difference between the acarbose and your capsules in terms of enzyme inhibition.

Response 5.4:   Section 3.2.2, it was added acarbose activity on the graphs (line 455).

Lines 383-392- again it would be useful to include data for acarbose on the graph

Response 5.5:   Section 3.2.2 it was added acarbose activity on the graphs (line 455).

Lines 399-401- sentence repeats your result from earlier, no need to add it here in my opinion

Response 5.6:  The sentence of 437-455 was changed and modified (lines 498-500).

Section 3.2.2- lacks discussion. What could explain your observations?

Response 5.7:  It was more explained in lines 445-452.

Section 3.2.3 (and 3.2.2.)- please comment on whether B. coagulans itself was found in any sources to have antidiabetic properties, and if yes, give specific mechanisms.

Response 5.8:  The sentence was added in lines 483-488

Section 3.3- no need to talk about previous sections here (lines 480-481),

Response 5.1:   It was modified in lines 552-556.

Please add a discussion on how effective were other microencapsulation materials used in the literature and compare with your findings (of course, the survival rate depends on the in-vitro digestion assay as well but quoting relative values to unencapsulated controls should give some good approximation). Also, it is important to add here a note on whether you would expect probiotics to be released at any digestion stage from your capsules. This was not mentioned anywhere in the manuscript

Response 5.1:   The paragraph was added detail in lines 523-531

Conclusions- Instead of re-stating your results, it would be better to summarise what you have found overall. From what I understood the key ingredient in your formulation is skimmed milk and depending on its concentration in the final formulation you can get better results in terms of survival and synbiotic bioactivity. In my opinion, this is the key information to convey in the conclusion rather than comparing SMC5 and SMC7. In addition, in lines 501-504, please rephrase the sentence. It is not your report that needs further research, but your developed synbiotic.

Response:  The paragraph was revised (lines 559-573).

o Informed consent statement- Is this not required for human trials only?

Response:  Examining the effectiveness of encapsulation and investigating the potential of B. coagulans in various biological activities among human subjects is a vital undertaking. In our upcoming research phase, we intend to conduct these activities with human participants to validate their effectiveness. This step is essential for verifying the benefits of encapsulation and B. coagulans in human biological processes.

o Figures- All details are well visible; please add acarbose control data where appropriate (see

earlier comments) and consider changing abbreviations SGI and SIF to full names in the figure

(should fit and be visible)

Response:  Figure 4 and Figure 6 were revised (lines 458 and 533)

Figure captions- for figures 1-4 you have included very long captions. In my opinion, there is no need to explain your formulations in captions, if you want you may only relate to the skimmed milk content and explain what the numbers mean, but then do this also in the caption for Figure 5. The p-value should be ≤ in relation to your significance level

Response:  The figure caption was modified per suggestion.

For Figure 5- the figure shows a logarithmic reduction so consider including this term instead of “survival decreased” which is not grammatically correct

Response:  The sentence was revised (line 533).

o References- the list contains current literature, most citations from the last 5 years, but authors could use some more references for the discussion of these numerous results.

Response:  The author has added references to make them more appropriate.
